# Preeclampsia Treatment Aspirin/Clampsilin: Oxidative Stress, sFlt-1/PIGF Soluble Tyrosine Kinase 1, and Placental Growth Factor Monitoring

**DOI:** 10.3390/ijms252413497

**Published:** 2024-12-17

**Authors:** Denitsa Kostadinova-Slavova, Kamelia Petkova-Parlapanska, Irina Koleva, Mariya Angelova, Rafaah Sadi J. Al-Dahwi, Ekaterina Georgieva, Yanka Karamalakova, Galina Nikolova

**Affiliations:** 1Obstetrics and Gynaecology Clinic, UMHAT “Prof. St. Kirkovich”, 6000 Stara Zagora, Bulgaria; denitsa.kostadinova@trakia-uni.bg (D.K.-S.); rafa.dzhasim@trakia-uni.bg (R.S.J.A.-D.); 2Department of Medical Chemistry and Biochemistry, Medical Faculty, Trakia University, 11 Armeiska Str., 6000 Stara Zagora, Bulgaria; kamelia.parlapanska@trakia-uni.bg (K.P.-P.); irina.d.koleva@trakia-uni.bg (I.K.); yanka.karamalakova@trakia-uni.bg (Y.K.); 3Department of Obstetrics and Gynecology, Medical Faculty, Trakia University, 11 Armeiska Str., 6000 Stara Zagora, Bulgaria; mariya.angelova@trakia-uni.bg; 4Department of General and Clinical Pathology, Forensic Medicine, Deontology and Dermatovenerology, Medical Faculty, Trakia University, 11 Armeiska Str., 6000 Stara Zagora, Bulgaria; ekaterina.georgieva@trakia-uni.bg

**Keywords:** pregnancy, preeclampsia, oxidative stress, NO, NOS

## Abstract

The present study aimed to investigate and compare oxidative stress biomarkers and antioxidant enzyme activity in the serum of women at risk of developing preeclampsia (PE) to prevent adverse pregnancy outcomes through early intervention. Changes in soluble fms-like tyrosine kinase-1 (sFlt-1) and placental growth factor (PlGF) levels were measured between 11 and 13 gestational weeks (gw.) before the onset of preeclampsia and its associated complications. This study evaluated the feasibility of the sFlt-1/PlGF biomarker ratio in predicting preeclampsia and adverse pregnancy outcomes, with the goal of preventive therapy with acetylsalicylic acid (150 mg daily), with acetylsalicylic acid (75 mg daily) and Clampsilin. For this purpose, the following were evaluated: (1) the levels of reactive oxygen species (ROS) and reactive nitrogen species (RNS) as parameters of oxidative stress; (2) lipid oxidation; (3) antioxidant enzyme activity; and (4) cytokine production. Analysis of the results showed that pregnant women at risk of preeclampsia had significantly higher levels of ROS, lipid oxidation, and superoxide anion radical (•O_2_^−^) levels compared to normal pregnancies. In PE, depleted levels of nitric oxide (NO), impaired NO synthase system (NOS), and reduced antioxidant enzyme activity (*p* < 0.03) suggest that PE patients cannot compensate for oxidative stress (OS). In conclusion, oxidative stress in PE plays a key role, which arises from placental problems and affects both mother and baby. The groups with acetylsalicylic acid therapy (150 mg and 75 mg) were better affected compared to those on Clampsillin.

## 1. Introduction

In preeclampsia, the invasion of fetal trophoblasts is impaired. Hence, the remodeling of the maternal spiral arteries is reduced, and as a result, blood flow and placental oxygen of the placenta are reduced [1,2]. To compensate for the lack of blood flow, the mother develops hypertension in the second or third trimester of pregnancy. The problems stop after birth, suggesting the placenta is the source of the problem [3]. The normal course of pregnancy involves variations in hemodynamics in which the placenta allows the exchange of nutrients and waste disposal between mother and fetus [4]. During the first trimester of pregnancy, the mother-fetus bond develops, and then extravillous trophoblasts from the placenta invade the maternal decidua [5]. During this time, the maternal spiral arteries from the decidualized endometrium remodel, upgrading from low-capacity, high-resistance vessels to high-capacity, low-resistance vessels [6]. The process is accompanied by the replacement of arterial smooth muscle and elastic tissue with fibrinoid material. As a result of PE, trophoblastic invasion and uteroplacental artery remodeling are exacerbated, increasing levels of reactive oxygen species (ROS), hypoxia, and endothelial dysfunction (ED) [7,8]. The physiology of pregnancy requires adequate placental oxygenation. High oxygen fluxes increase ROS levels, leading to the appearance of markers of oxidative stress. For this purpose, we selected women at risk of developing preeclampsia to prevent adverse pregnancy outcomes through early intervention [9]. Changes in soluble fms-like tyrosine kinase-1 (sFlt-1) and placental growth factor (PlGF) levels were measured between 11–13 g.w. before the onset of preeclampsia and its associated complications. This study evaluated the feasibility of the sFlt-1/PlGF biomarker ratio in predicting preeclampsia and adverse pregnancy outcomes, with the goal of preventive therapy with acetylsalicylic acid (150 mg daily), acetylsalicylic acid (75 mg/daily) and Clampsillin.

## 2. Results

### 2.1. Levels of NO, eNOS, and iNOS

The Normotensive Pregnant (NP) group showed significantly higher levels of nitric oxide radicals (NO•) compared to the Preeclamptic pregnant women group (*p* = 0.00) (Figure 1A) and the group with Aspirin therapy 150 mg/day (*p* = 0.003), and Clampsilin (*p* = 0.002) group. The eNOS (Figure 1B) and iNOS (Figure 1C) concentrations were significantly lower in the PE group compared to the groups with Aspirin and Clampsillin therapy. The results for the group with Aspirin 75 mg (*p* = 0.06) were statistically insignificant lower than the NP group.

### 2.2. Oxidative Stress Assessment by Measuring the MDA, ROS and •O_2_^−^

The PE group showed higher levels of malondialdehyde (MDA) compared to the NP group (*p* = 0.03) and groups with therapy Aspirin 150 mg/day (*p* = 0.001), Clampsillin (*p* = 0.02), and Aspirin 75 mg/day (*p* = 0.00) (Figure 2A).

ROS levels (Figure 2B) and serum •O_2_^−^ levels (Figure 2C) were also increased in the PE group compared to the NT group and groups with therapy. In all studied markers, the results of the group with therapy Aspirin 75 mg/day are close to the NP group.

### 2.3. The Levels of Pro-Inflammatory Cytokines

In the PE group, the interleukin 6 (IL-6) levels (Figure 3A) were statistically significantly increased both compared to NP (LSD post hoc test, *p* < 0.05) and compared to groups with therapy—Aspirin 150 mg/day (LSD post hoc test, *p* < 0.05) and Clampsilin (LSD post hoc test, *p* < 0.05). In the group with the therapy of Aspirin 75 mg/day (LSD post hoc test, *p* < 0.05), the results were close to the NP (LSD post hoc test, *p* = 0.1).

The mean value of tumor necrosis factor (TNF-α; Figure 3B) in the PE group was statistically significantly higher compared to the NP (mean 98.99 ± 10.78 pg/mL vs. mean 39.89 ± 11.12 pg/mL, LSD post hoc test, *p* < 0.05), and to groups with therapy (Aspirin 150 mg/day mean 98.99 ± 10.78 pg/mL vs. mean 40.11 ± 8.81 pg/mL; Clampsilin mean 98.99 ± 10.78 pg/mL vs. mean 42.95 ± 6.74 pg/mL, LSD post hoc test, *p* < 0.05). The statistically insignificant lower level of TNF-α in a group with Aspirin 75 mg/day therapy was compared to the NP group (mean 44.05 ± 5.31 pg/mL vs. mean 39.89 ± 11.12 pg/mL, *p* = 0.06). The interferon gamma results (ITF-γ; Figure 3C) showed a statistically significant increase in the PE group to NP (*p* < 0.05) and against both groups with the therapy of Aspirin 150mg/day (*p* < 0.05) and Clampsilin (*p* < 0.05). Close to the NP group were the results from the Aspirin 75 mg therapy group to NP (*p* < 0.1).

Transforming growth factor-β (TGF-β; Figure 3D) also showed a statistically significant increase in the PE group compared to NP (*p* < 0.05) and against groups with therapy: Aspirin 150 mg/day *p* < 0.05 and Clampsilin (*p* < 0.05). Interleukin 1α (IL-1α; Figure 3E) level in the PE women was statistically significantly higher compared to NP (LSD post hoc test, *p* < 0.05) and groups with therapy (Aspirin 150 mg/day (*p* < 0.05) and Clampsilin (*p* < 0.05)). Interleukin 1β (IL-1β; Figure 3F), Interleukin 17β (IL-17; Figure 3G) and Interleukin 22 (IL-22, Figure 3H) were not shown statistically difference between the PE group and NP and therapy groups.

For the group with Aspirin 75 mg/day therapy, the results for TGF-β (Figure 3D), IL-1α (Figure 3E), IL-1β (Figure 3F), IL-17 (Figure 3G) and IL-22 (Figure 3H) were almost as the NP group.

## 3. Discussion

Aspirin is one of the oldest and most widely used drugs. Low-dose Aspirin therapy in high-risk pregnancies is associated with a reduction in the incidence of PE and subsequent complications [10,11]. In recent years, it has been reported that low-dose Aspirin, initiated at or before 16 weeks of gestation, is associated with a strong reduction in the overall risk of PE and a significant reduction in premature PE [12,13]. First-trimester Aspirin use is not associated with an increased risk of fetal abnormalities, and there is no evidence of increased maternal bleeding or placental abruption [12,14]. Furthermore, no association has been found between low-dose Aspirin use in the third trimester and antenatal closure of the ductus arteriosus, intraventricular hemorrhage, or neonatal bleeding [15]. The imbalance of thromboxane and prostacyclin in preeclampsia suggests the need to assess the dose of Aspirin. Wallenburg et al. [16] conducted a randomized, placebo-controlled, double-blind trial of 60 mg/day Aspirin in 46 pregnant women at risk of preeclampsia. Twelve of 23 pregnant women taking a placebo developed preeclampsia, while only 2 of 21 pregnant women taking Aspirin developed preeclampsia [16]. The results showed that a low dose of 60 mg/day Aspirin reduced the incidence of preeclampsia and led to a balance between thromboxane/prostacyclin [17]. High doses of Aspirin 150 mg suppress the production of prostaglandins and thromboxane and inhibit inflammation and platelet aggregation. Ghesquiere et al. [18] describe a randomized meta-study that a dose of Aspirin ≥ 150 mg/day was significantly reduced in premature preeclampsia compared to an Aspirin dose of ≤75 mg [18,19,20,21].

The results presented here report patients at high and moderate risk of preeclampsia, stratified according to the analysis of sFlT-1/PIGF, sFlT-1, and PIGF and treated. The group of pregnant women at moderate risk of PE was divided into two subgroups according to therapy: one was on Clampsillin therapy (2 sachets), and the second was on low-dose Aspirin (75 mg/day).

Oxidative stress induces the adhesion of leukocytes and platelets to the endothelium, which increases the release of cytokines and antiangiogenic factors (Figure 3A–H). ROS plays a critical role in endothelial dysfunction implicated in PE. In normal pregnancy, NO contributes to the maintenance of vascular tone, increases uterine blood flow, and mediates endothelium-dependent vasodilation. Endothelial cells produce NO, which is a potent vasorelaxant and anticoagulant factor [22,23,24]. Activation of NO synthase (NOS) triggers the production of NO during pregnancy (Figure 1), and the observed endothelial dysfunction is a direct result of low NO levels, a likely cause of PE [25,26]. The increased ROS production suppresses the expression of NOS enzymes, reducing the bioavailability of NO (Figure 1). In addition to ROS and NO, peroxynitrite (ONOO^−^) is formed, which oxidizes important biomacromolecules, blocks important vascular signaling pathways, disrupts the redox balance, and causes endothelial dysfunction [27,28]. Antioxidant enzymes (SOD, GSH-Px, and CAT) perform an important function in detoxification and protection against endogenous and exogenous oxidants [29,30]. A close relationship has been established between the severity of lipid damage and the clinical severity of preeclampsia. In preeclampsia, the body’s antioxidant capacity is reduced, and the body fails to maintain redox balance, which leads to lipid peroxidation, compared to normotensive pregnant women [31]. There are conflicting data on the activity of GPx in relation to the severity of PE. Freire et al. [32] reported that no statistical difference was found in the activity of GPx and CAT compared to controls, as well as significant differences in the activity of CAT in different severity of PE [32].

According to data, CAT and GPx activity is increased in PE compared to NB [33,34]. A systematic review by Taravati and Tohidi reported that GPx and CAT activity in serum or plasma was higher in PE than in normotensive pregnant women. It is likely that GPx and CAT activity depend on the levels of lipid peroxides and H_2_O_2_ in the body in PE-pregnant women. From the results presented here (Table 1), a statistical relationship was observed between GPx levels in PE and NB, with the activity being higher in normal pregnancy. An inversely proportional relationship was reported in CAT activity, with the enzyme levels being higher in PE pregnant women. The enzyme activity of endogenous antioxidant enzymes increased in the groups of patients undergoing therapy with Aspirin and Clampsilin (*p* = 0.0), and no statistically significant difference was observed between patients treated with Aspirin and Clampsilin (Table 1). Patients at high risk of PE were treated with 150 mg/day Aspirin.

The results showed that the Aspirin-treated group (75 mg/day) responded better to the treatment (Section 2). A similar effect was observed in the group at high risk of preeclampsia and treated with 150 mg/day Aspirin. Clampsilin contains a protein and regulates vesicular trafficking, clathrin-mediated endocytosis, and signal transduction pathways. It is also involved in receptor internalization and nutrient absorption. The reported results show that for the prevention of high-risk PE, the dose of 150 mg/day aspirin is preferable to the standard 75 mg/day Aspirin. The 75 mg/day dose of Aspirin affects moderately severe PE better and is preferable to treatment with Clampsilin. Most studies have reported that women with PE have higher blood concentrations of cytokines (IL-6, IL-1α, TNF-α, and CRP) compared with women with uncomplicated pregnancies [34]. Patients with PE have elevated cytokine levels. Levels of interferon-gamma (IFN-γ), interleukins IL-1 (alpha and beta), IL-17, and IL-22 play a crucial role in the development of PE.

The inflammatory cytokine IL-17 is produced by activated T cells and stimulates epithelial, endothelial cells, and fibroblasts to produce various cytokines that increase inflammation [34]. As a member of the IL-10 family of proinflammatory cytokines, the main function of IL-22 is to maintain and restore epithelial integrity. Elevated levels of IL-22 activate receptors, especially angiotensin type 1, increase oxidative stress and vasoconstriction and thereby increase blood pressure in pregnant women [35].

Patients with PE showed a statistically significant increase in circulating IFN-γ levels (Figure 3C) compared to controls and Aspirin and Clampsilin 150 mg/day treatment groups. All results in the 75 mg/day Aspirin group were similar to the NP group. Autoimmune and autoinflammatory diseases increase IFN-γ expression. A statistical difference in IL-1α levels was observed in PE compared to NB (Figure 3E), which is probably due to an increased inflammatory response with increasing gestational age. No statistically significant difference was observed between the PE group and normotensive pregnancy cytokines IL-1 beta, IL-17, and IL-22 (Figure 3F–H), as well as when comparing the groups of patients undergoing Aspirin and Clampsilin therapy.

## 4. Materials and Methods

### 4.1. Chemicals

All reagents were analytical grade (N-tert-butyl-α-phenylnitrone), PBN, carboxy-PTIO potassium salt, dimethyl sulfoxide DMSO, 2-thiobarbituric acid and were purchased from Merck, Sigma-Aldrich, Sofia, Bulgaria EAD.

### 4.2. Diagnostic Criteria for Outcomes

Preeclampsia is a multisystem syndrome developing in the second half of pregnancy. It is characterized by hypertension, proteinuria, and edema in the absence of other dysfunction of the maternal organs. The diagnosis of PE should be based on the recommendations of ACOG, 2013, in which the systolic pressure is >140 mm Hg and/or the diastolic blood pressure is ≥90 mmHg and develops after 20 weeks of gestation in previously normotensive women on at least two occasions with an interval of 4 h. There is also preeclampsia, which is superimposed on chronic hypertension (hypertension present before conception or established hypertension before the 20th week of gestation) and which is accompanied by proteinuria or dysfunction of the maternal organism. Preeclampsia occurs in 3% to 8% of nulliparas and 1% to 3% multiparas, with one-third of cases leading to delivery before 37 completed weeks of gestation (early onset preeclampsia) and two-thirds of delivery occurring after 37 weeks (late-onset preeclampsia). Proteinuria is diagnosed with the following criteria: Proteinuria is the presence of >300 mg of protein in a 24-h urine collection, or the ratio of urine protein to creatinine is >30 mg/mmol.

Preeclampsia is one of the causes of maternal and perinatal mortality. The most serious complications that can lead to maternal death are eclampsia (seizures in a woman with severe preeclampsia), cerebral hemorrhage or stroke, disseminated intravascular coagulation (DIC), and HELLP syndrome (hemolysis, elevated liver enzymes, and low platelets). Other severe complications include brain edema, blindness, kidney failure, liver failure, or pulmonary edema.

Preeclampsia is associated with reduced blood supply to the placenta with subsequent impairment of fetal growth and an increased risk of stillbirth. In addition, a high percentage of women with preeclampsia require preterm delivery due to maternal and/or fetal indications. Newborns are exposed to additional risks arising from prematurity. These include neonatal death, cerebral hemorrhage, difficulty breathing and feeding, jaundice, retinopathy, and prolonged hospitalization and disability.

### 4.3. Prediction of Preeclampsia

Fetal medicine specialists approach screening for preeclampsia by using a combination of the features of the pregnancy and medical history with the results of various combinations of biophysical and biochemical measurements taken at different times during pregnancy. The combined screening of maternal factors, MAP (mean arterial pressure), UTPI (uterine artery pulsatility index), PLGF, PAPP–A, and sFLT-1 predicted all cases of early preeclampsia (<34 weeks). Serum sFLT1 measured at 12 weeks improves screening performance for early preeclampsia but not for preeclampsia >34 weeks. Screening by maternal factors (MAP, UTPI, PLGF, PAPP-A) predicted 85% of premature preeclampsia (<37 weeks) and 45% of late preeclampsia (>37 weeks). In pregnancies that develop preeclampsia, MAP, UTPI, and sFLT-1 values increase, and PLGF decreases.

*Inclusion criteria are:* pregnant women with a gestational age of 11 to 34 weeks, with the gestational age being dated according to the first day of the LMP, singleton pregnancies, spontaneous pregnancies, live fetuses, pregnant women over 18 years of age, without systemic autoimmune diseases, without established aneuploidy of the fetuses (by triple test, performed in the period 11 to 13 weeks + 6 days).

*Exclusion criteria are:* presence of gestational diabetes, diabetes mellitus type 1 and 2, presence of systemic autoimmune disease, in-vitro fertilization, established aneuploidy or other fetal abnormalities, termination of pregnancy due to fetal death or abortion.

### 4.4. Patients

Pregnant women (n = 111) aged 28 ± 8 years visiting the outpatient department were also examined after 12 gestational weeks (for CRL/Crown-rump length) (41–79 mm) sFlT-1/PIGF results; sFlT-1 soluble tyrosine kinase; PIGF placental growth factor showed high and intermediate risk of preeclampsia (n = 77) and met the inclusion criteria were included in the study after written informed consent. The study was conducted over 1 year, and informed consent was obtained from all subjects involved.

Medium-risk (n = 43) patients were divided into two groups: the first (n = 21) on Clampsillin therapy (2 sachets daily, L-arginine, choline bitartrate, maltodextrin, natural color (beet, betanin), lutein, strawberry flavor, emulsifier (silica), sweeteners (sucralose, stevia), natural flavor, citric acid, and vitamin D)), and the second (n = 22) on low-dose Aspirin Acard (75 mg/daily). The group at high risk of preeclampsia (n = 34) was on 150 mg daily Aspirin Acard (150 mg gastro-resistant tablets). All results were compared in pregnant women with normal pregnancies and those obtained before taking Aspirin and Clampsillin.

The therapy is used for a period of 14 g.w. and continues daily for 3 days before the estimated day of delivery. The biomarkers of oxidative stress are measured 8 weeks after the onset of therapy (24–26 g.w.).

### 4.5. Blood Samples Preparation

Fasting samples of venous blood from each patient and controls were collected in the morning between 07:00 and 09:00 h in clot blood tubes, centrifuged at 3000 rpm for 10 min at 5 °C, and serum was carefully separated and used immediately.

### 4.6. Electron Paramagnetic Resonance (EPR) Study

All EPR measurements were performed at room temperature on a Bruker BioSpin GmbH, Ettlingen, Germany, equipped with a standard resonator. All EPR experiments were carried out in triplicate and repeated thrice. Spectral processing was performed using Bruker WIN-EPR accessed 2019 and Simfonia, software version 1.2. 

### 4.7. Evaluation of the ROS Product Levels

The levels of ROS were determined following Shi et al. [36] with some modifications. To investigate in real time the formation of ROS in the sera of patients and controls, ex vivo EPR spectroscopy was used combined with N-tert-butyl-alpha-phenylnitrone (PBN) as a spin-trapping agent. PBN, upon reaction with unstable radicals, such as ROS, forms relatively stable spin adducts that can be subsequently detected by EPR spectroscopy. The EPR settings were as follows: 3503.73 G center field, 20.00 mW microwave power, 5 G modulation amplitude, 50 G sweep width, 1 × 10^5^ gain, 81.92 ms time constant, 125.95 s sweep time, 5 scans per sample.

### 4.8. Evaluation of the •NO Radical Levels

Based on the methods published by Yoshioka et al. [37] and Yokoyama et al. [38], we developed and adapted the EPR method for the estimation of •NO radical levels. Briefly, imidazoline-1-oxyl-3-oxide potassium salt (Carboxy PTIO.K) dissolved in a mixture of 50 mMTris (pH 7.5) and DMSO in a ratio 9:1 was added to a 50 μM solution of carboxy 2-(4-carboxyphenyl) -4, 4, 5, 5 –tetramethyl. The EPR settings were as follows: 3505 G centerfield, 6.42 mW microwave power, 5 G modulation amplitude, 75 G sweep width, 2.5 × 10^2^ gain, 40.96 ms time constant, 60.42 s sweep time, with 1 scan per sample.

### 4.9. Superoxide Anion Radical •O_2_^−^

The serum was incubated for 30 min at 37 °C with a superoxide-sensitive spin probe, 1-hydroxyl-3-methoxycarbonyl-2, 2, 5, 5-tetramethylpyrrolidine (CMH), and immediately measured. The amplitude of the spectrum is directly proportional to the *•O*_2_^−^ concentration [39].

### 4.10. Enzyme-Linked Immunosorbent Assay

All markers of oxidative stress were measured with ELISA kits following the manufacturer’s instructions.

### 4.11. Statistical Analysis

Statistical analysis was performed with Statistica 8, StaSoft, Inc. (Madrid, Spain), and the results were expressed as means ± S.E. All data were expressed as means ± SE and obtained by one-way ANOVA, and in the LSD post hoc test, *p* > 0.05 was considered statistically significant. LSD post hoc tests were used to define which groups were different from each other.

## 5. Conclusions

In PE, oxidative stress, arising from placental problems, plays a key role and affects both mother and baby. The role of ROS/RNS production in PE is still controversial, and there is still no consensus on whether •NO levels are high or low during disease development. Spectrophotometric studies have reported that plasma and serum •NO levels are higher in preeclamptic women than in normotensive women. However, studies conducted with state-of-the-art •NO detection technologies have shown the opposite. Also, performed in placental cells, tissue, and umbilical cord blood at the placental/fetal stage, NO levels were reported to be lower in PE. The generation of superoxide in PE is very well established. Changes in placentation during the first trimester lead to the development of OS and dysfunction of the vascular endothelium, which plays a key role in the development of pregnancy complications such as preeclampsia. High- and low-dose Aspirin therapy is more effective than the protein product Clampsillin.

## Figures and Tables

**Figure 1 ijms-25-13497-f001:**
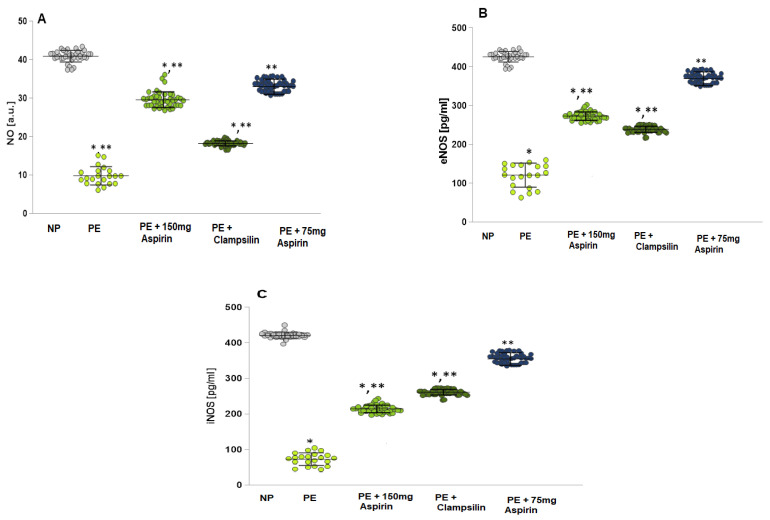
Present the NO levels, eNOS, and iNOS in serum samples. (**A**) NO: Normotensive pregnancy (NP); PE patients; PE + 150 mg/day Aspirin; PE + Clampsilin; PE + Aspirin 75 mg/day. (**B**) eNOS: NP; PE patients; PE + 150 mg/day Aspirin; PE + Clampsilin; PE + Aspirin 75 mg/day. (**C**) iNOS: NP; PE patients; PE + 150 mg/day Aspirin; PE + Clampsilin; PE + Aspirin 75 mg/day. LSD post hoc test; * *p* < 0.05 vs. NP group; ** *p* < 0.05 vs. PE group.

**Figure 2 ijms-25-13497-f002:**
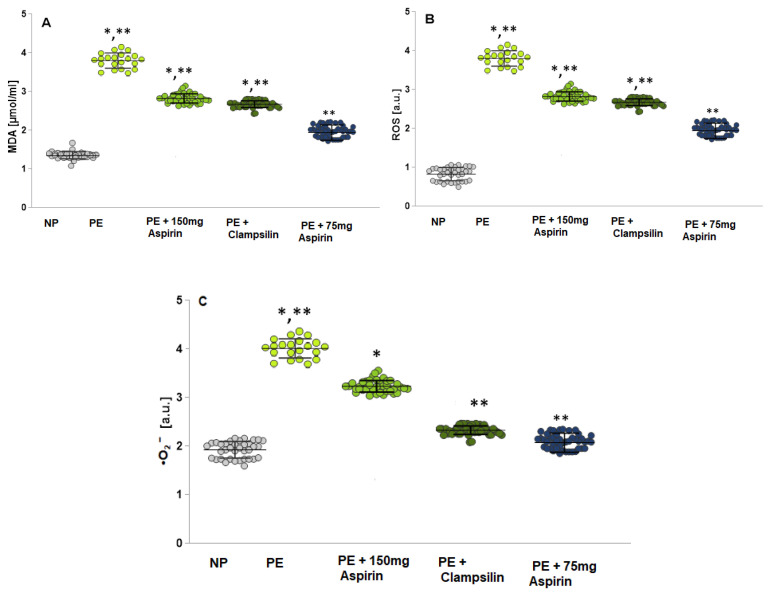
The levels of oxidative stress markers are presented as MDA, ROS production, and •O_2_^−^. (**A**) MDA levels -NP; PE; PE + Aspirin 150mg/day; PE + Clampsilin; PE + Aspirin 75 mg/day. (**B**) ROS production— NP; PE; PE + Aspirin 150mg/day; PE + Clampsilin; PE + Aspirin 75 mg/day. (**C**) NP; PE; PE + Aspirin 150 mg/day; PE + Clampsilin; PE + Aspirin 75 mg/day. LSD post hoc test; * *p* < 0.05 vs. NP group; ** *p* < 0.05 vs. PE group.

**Figure 3 ijms-25-13497-f003:**
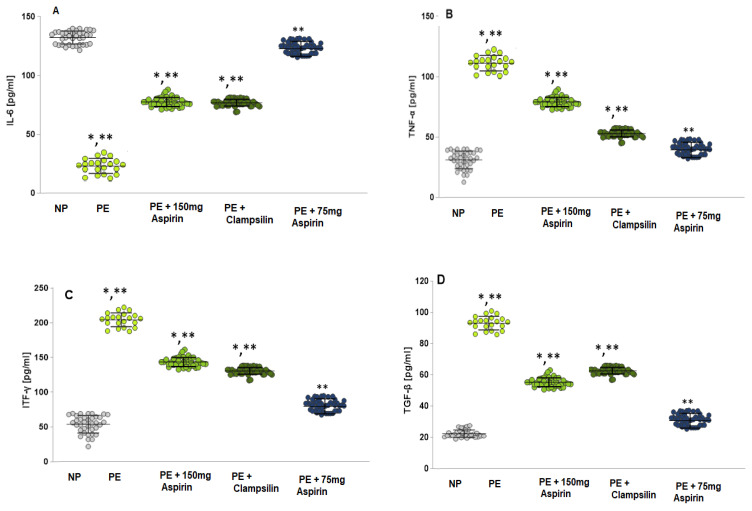
Pro-inflammatory cytokine levels: (**A**) IL-6; (**B**) TNF-α; (**C**) IFN-γ; (**D**) TGF-β; (**E**) IL-1α; (**F**) IL-1β; (**G**) IL-17; (**H**) IL-22; LSD post hoc test, * *p* < 0.05 vs. NP; ** *p* < 0.05 vs. PE.

**Table 1 ijms-25-13497-t001:** Characteristics of women included in the trial (The sFlt-1/PlGF ratio of 38–85 indicates early onset PE) or 38–110 (late onset PE. The sFlt-1/PlGF ratio provides information before the onset of overt signs and symptoms. An sFlt-1/PlGF ratio of 38–85 or 38–110 provides additional information about which women are at moderate or high risk of developing PE within 4 weeks. Current PE or placental-related disorders can be excluded, but women are at (high) risk (especially in the early onset group)).

Variables	Normotensive Pregnant (NP) n = 34	PE Before Therapy (n = 77)	PE with Aspirine Acard (n = 34) 150 mg/day	PE with Clampsilin Therapy (n = 21) 2 Sachets Daily	PE with Aspirine Acard (n = 22) 75 mg/day
Age at delivery (years)	27.33 ±9.28	29.17 ± 10.01	30.17 ± 10.01	28.17 ± 9.25	28.11 ± 10.36
BMI (kg/m^2^) mean ± SD	24.5 (21.9–28.4)	27.1 (23.5–32.3)	33.56 (31.4 ± 36.4)	33.76 (31.6 ± 36.1)	33.61 (31.4 ± 36.9)
Gestational age at sampling (weeks)	11–13^+6^	11–13^+6^	20–34	20–34	20–34
Blood shugar mmol/L, mean ±SD	4.98 ± 0.51	5.12 ± 2.35	5.32 ± 4.89	4.85 ± 5.41	4.98 ± 5.02
Chronic hypertension	no	no	no	no	no
HbA1_C_%	5.69 ± 0.44	5.08 ± 0.52	5.16 ± 0.27	5.27 ± 0.11	5.11 ± 0.18
Diabetes Mellitus (DM) Status	no	no	no	no	no
sFlT-1/PIGF	5.03 ± 4.62	87.61 ± 14.11	70.85 ± 10.04	40.07 ± 9.33	72.01± 6.07
sFlT-1 (soluble thyrosinkinase)	991 ± 1.17	5005 ± 123.22	4002 ± 111.34	4208 ± 174.01	4199 ± 169.22
PIGF (placental growth factor)	197 ± 15.05	90 ± 12.85	103 ± 18.01	105 ± 14.15	98 ± 10.01
blood pressure	110/60	145/100	120/85	115/80	115/80
proteinuria (UPCR) mg/mmol	<30	>45	<40	<40	<40
Superoxide dismutase SOD (U/gHb)	97.13 ± 12.78	29.41 ± 2.99	59.77 ± 7.81	61.01 ± 8.05	58.55 ± 10.13
Catalase CAT (U/gHb)	42.71 ± 9.09	113.21 ± 21.12	72.66 ± 13.45	74.68 ± 13.55	65.47 ± 5.16
Glutathione peroxidase GSH-Px (U/gHb)	278.44 ± 31.25	101.59 ± 33.55	147.13 ± 41.25	151.71 ± 31.13	189.87 ± 7.61

## Data Availability

The data presented in this study are available on request from the corresponding author.

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
