# Peer review of "Preeclampsia Treatment Aspirin/Clampsilin: Oxidative Stress, sFlt-1/PIGF Soluble Tyrosine Kinase 1, and Placental Growth Factor Monitoring"

_ijms, 2024, doi:10.3390/ijms252413497_

Round 1
Reviewer 1 Report
Comments and Suggestions for Authors
Thank you for your kind invitation to review this manuscript.
This is an interesting and solid work in which the authors have prepared a summary very comprehensive on the possible biomarkers of elevated oxidative stress in pregnant patients suffering from preeclampsia. The conclusions are based on the results obtained on the possible oxidative stress-targeted therapy improving risk of preeclampsia.
The only thing I would change would be to complete acronyms that have not been previously indicated, such as ROS in section abstract instead of section 1 introduction, •O2‾ in section 2.2 Oxidative Stress assessment by measuring the MDA, ROS and •O2‾ or OS in section 3 discussion.
Author Response
RESPONSES TO THE Reviewers COMMENTS
Thank you very much. We appreciate reviewers’ comments. All corrections in the manuscript are highlighted in green.
Reviewer
Thank you for your kind invitation to review this manuscript
This is an interesting and solid work in which the authors have prepared a summary very comprehensive on the possible biomarkers of elevated oxidative stress in pregnant patients suffering from preeclampsia. The conclusions are based on the results obtained on the possible oxidative stress-targeted therapy improving risk of preeclampsia.
Point 1
The only thing I would change would be to complete acronyms that have not been previously indicated, such as ROS in section abstract instead of section 1 introduction, •O2‾ in section 2.2 Oxidative Stress assessment by measuring the MDA, ROS and •O2‾ or OS in section 3 discussion.
Thank you very much
Answer: Done
Reviewer 2 Report
Comments and Suggestions for Authors
Congratulations to the authors for addressing this interesting and important issue.
However, there are certain things that need to be clarified and restuctured. The major part of the discussion, except the first paragraph, should be in the introduction section. In the discussion other studies should be elaborated along with the results from the present study. It was stated "The majority of studies used relatively low doses of 5-100 mg aspirin [12], and a few reported using 150 mg/day [14]" Authors should mention some of these studies and discuss according to the results.
Lines 109-111 "In PE group the interleukin 6 (IL-6) levels (Figure 3, A) were statistically significantly increased both compared to NP (LSD post hoc test, p<0.05) and compared to groups with therapy." However, the graph shows the opposite results where the lowest IL6 is in PE group?
What was the criteria for classifying PE into high or medium risk?
Why were 89 patients included in the study, how was the sample size calculated?
How long did the patients take medications and when were the parameters measured?
Please elaborate more on CAT activity and glutathione levels in lines 209-212 and add more references since the data are controversial.
Author Response
RESPONSES TO THE Reviewers' COMMENTS
We appreciate reviewers’ comments. All corrections in the manuscript are highlighted in yellow.
Reviewer
Congratulations to the authors for addressing this interesting and important issue.
However, there are certain things that need to be clarified and restuctured. The major part of the discussion, except the first paragraph, should be in the introduction section. In the discussion other studies should be elaborated along with the results from the present study. It was stated "The majority of studies used relatively low doses of 5-100 mg aspirin [12], and a few reported using 150 mg/day [14]" Authors should mention some of these studies and discuss according to the results.
Point 1:
Lines 109-111 "In PE group the interleukin 6 (IL-6) levels (Figure 3, A) were statistically significantly increased both compared to NP (LSD post hoc test, p<0.05) and compared to groups with therapy." However, the graph shows the opposite results where the lowest IL6 is in PE group?
Answer 1: Thank you very much! It was our mistake and we corrected it:
In the PE group, the interleukin 6 (IL-6) levels (Figure 3, A) were statistically significantly increased both compared to NP (LSD post hoc test, p<0.05) and compared to groups with therapy—Aspirin 150 mg/day (LSD post hoc test, p<0.05) and Clampsilin (LSD post hoc test, p<0.05). In the group with the therapy of Aspirin 75 mg/day (LSD post hoc test, p<0.05), the results were close to the NP (LSD post hoc test, p=0.1).
Point 2: What were the criteria for classifying PE into high or medium risk?
Answer 2: The criteria for classifying PE were the blood pressure and the levels of proteinuria (Urine Protein creatinine ratio, UPCR) included in Table 1.
Point 3: Why were 89 patients included in the study, how was the sample size calculated?
Answer 3: Thank you very much! We corrected the mistake!
Pregnant women (n = 111) aged 28 ± 8 years visiting the outpatient department were also examined after 12 gestational weeks (for CRL/Crown-rump length) (41–79 mm) sFlT-1/PIGF results; sFlT-1 soluble tyrosine kinase; PIGF placental growth factor showed high and intermediate risk of preeclampsia (n = 77) and met the inclusion criteria were included in the study after written informed consent. The study was conducted over 1 year, and informed consent was obtained from all subjects involved. Medium-risk (n = 43) were divided into two groups: the first (n = 21) on Clampsillin therapy (2 sachets daily, L-arginine, choline bitartrate, maltodextrin, natural color (beet, betanin), lutein, strawberry flavor, emulsifier (silica), sweeteners (sucralose, stevia), natural flavor, citric acid, and vitamin D)), and the second (n = 22) on low-dose Aspirin Acard (75 mg/daily). The group at high risk of preeclampsia (n = 34) was on 150 mg daily Aspirin Acard (150 mg gastro-resistant tablets). All results were compared in pregnant women with normal pregnancies and those obtained before taking aspirin and clampsillin.
Point 4: How long did the patients take medications and when were the parameters measured?
Answer 4: The therapy is used for the period of 14 g.w. and continue daily until 3 days before the estimated day of delivery. The biomarkers of oxidative stress are measured like 8 weeks after the onset of therapy (24-26 g.w.).
Point 5: Please elaborate more on CAT activity and glutathione levels in lines 209-212 and add more references since the data are controversial.
Answer 5:
Antioxidant enzymes (SOD, GSH-Px, and CAT) perform an important function in detoxification and protection against endogenous and exogenous oxidants [29, 30]. A close relationship has been established between the severity of lipid damage and the clinical severity of preeclampsia. In preeclampsia, the body's antioxidant capacity is reduced and the body fails to maintain redox balance, which leads to lipid peroxidation, compared to normotensive pregnant women [31]. There are conflicting data on the activity of GPx in relation to the severity of PE. Freire et al. [32] reported that no statistical difference was found in the activity of GPx and CAT compared to controls, as well as significant differences in the activity of CAT in different severity of PE [32].
According to data, CAT and GPx activity is increased [33] in PE compared to NB. In a systematic review by Taravati and Tohidi, it was reported that GPx and CAT activity in serum or plasma was higher in PE than in normotensive pregnant women. It is likely that GPx and CAT activity depend on the levels of lipid peroxides and H2O2 in the body in PE-pregnant women. From the results presented here (Table 1), a statistical relationship was observed between GPx levels in PE and NB, with the activity being higher in normal pregnancy. An inversely proportional relationship was reported in CAT activity, with the enzyme levels being higher in PE pregnant women. The enzyme activity of endogenous antioxidant enzymes increased in the groups of patients undergoing therapy with aspirin and clampsilin (p = 0.0), and no statistically significant difference was observed between patients treated with aspirin and clampsilin (Table 1). Patients at high risk of PE were treated with 150 mg/day aspirin.
Reviewer 3 Report
Comments and Suggestions for Authors
This prospective study investigated the effects of two does (150 and 75 mg per day) of aspirin vs. Clampsilin in ROS and RNS as well as cytokine production in high-risk patients of preeclampsia.
While large confirmative data of what already were known in the literature are presented in this study, the prospective study design and comparisons of three groups do offer important clinically relevant evidence very much worthy of publishing.
In general, the paper for the most part is well written and data are properly interpreted. However, two major clarifications are needed.
First, how the blood samples are prepared? serum or plasma?
The comparisons of 75 vs. 150 mg per day were briefly discussed. This needs more data to justify.
Minor: a few types need to be corrected.
oxidative stress biomarkers and an- 19 tioxidant enzyme activity in the serum of women at risk of developing preeclampsia (PE), to pre- 20 vent adverse pregnancy outcomes through early intervention. Changes in soluble fms-like tyrosine 21 kinase-1 (sFlt-1) and placental growth factor (PlGF) levels were measured between 11 and 13 ges- 22 tational weeks (g.w.) before the onset of preeclampsia and its associated complications. This study 23 evaluated the feasibility of the sFlt-1/PlGF biomarker ratio in predicting preeclampsia and adverse 24 pregnancy outcomes, with the goal of preventive therapy with acetylsalicylic acid (150 mg daily), 25 with acetylsalicylic acid (75 mg daily) and Clampsilin. For this purpose, the following were evalu- 26 ated: (1) the levels of reactive oxygen species and reactive nitrogen species as parameters of oxida- 27 tive stress; (2) lipid oxidation; (3) antioxidant enzyme activity; and (4) cytokine production. 28 Analysis of the results showed that pregnant women at risk of preeclampsia had significantly 29 higher levels of ROS, lipid oxidation, and superoxide anion radical levels compared to normal 30 pregnancies. In PE, depleted levels of nitric oxide (NO), impaired NO synthase system (NOS), and 31 reduced antioxidant enzyme activity (p<0.03) suggest that PE patients cannot compensate for oxi- 32 dative stress. In conclusion, oxidative stress in PE plays a key role, which arises from placental 33 problems and affects both mother and baby. The groups with acetylsalicylic acid therapy (150mg 34 and 75 mg) were better affected compared to those on Clampsillin
Author Response
RESPONSES TO THE Reviewers' COMMENTS
We appreciate reviewers’ comments. All corrections in the manuscript are highlighted in blue.
Reviewer
This prospective study investigated the effects of two does (150 and 75 mg per day) of aspirin vs. Clampsilin in ROS and RNS as well as cytokine production in high-risk patients of preeclampsia. While large confirmative data of what already were known in the literature are presented in this study, the prospective study design and comparisons of three groups do offer important clinically relevant evidence very much worthy of publishing. In general, the paper for the most part is well written and data are properly interpreted.
However, two major clarifications are needed.
Point 1: First, how the blood samples are prepared? Serum or plasma?
Answer: Done
Blood samples preparation
Fasting samples of venous blood from each patient and controls were collected in the morning between 07:00 and 09:00 h, in clot blood tubes, centrifuged at 3,000 rpm for 15 min, and serum was carefully separated and used immediately.
Point 2: The comparisons of 75 vs. 150 mg per day were briefly discussed. This needs more data to justify. Minor: a few types need to be corrected.
Answer 2: Done
Round 2
Reviewer 2 Report
Comments and Suggestions for Authors
Authors answered to all requests appropriately.
I only have one doubt. What are the cut off values of sFlt-1/PlGF for PE classification, whether it is medium or high risk?
Author Response
Thank you for your helpful and constructive note!
Comment: What are the cut off values of sFlt-1/PlGF for PE classification, whether it is medium or high risk?
Answer: The sFlt-1/PlGF ratio of 38–85 indicates early onset PE) or 38–110 (late onset PE). The sFlt-1/PlGF ratio provides information before the onset of overt signs and symptoms. An sFlt-1/PlGF ratio of 38–85 or 38–110 provides additional information about which women are at moderate or high risk of developing PE within 4 weeks. Current PE or placental-related disorders can be excluded, but women are at (high) risk (especially in the early-onset group).